# Differences in Sleep Offset Timing between Weekdays and Weekends in 79,161 Adult Participants in the UK Biobank

**Rachael M. Kelly** [1], **John H. McDermott** [2] **and Andrew N. Coogan** [1,3,*]

1   Department of Psychology, Maynooth University, W23 X021 Maynooth, Ireland
2   Academic Department of Endocrinology, Royal College of Surgeons in Ireland, Connolly Hospital Blanchardstown, D15 X40D Dublin, Ireland
3   Kathleen Lonsdale Institute for Human Health Research, Maynooth University, W23 X021 Maynooth, Ireland
*   Correspondence: andrew.coogan@mu.ie; Tel.: +353-17086624

**Abstract:** Variability in the timing of daily sleep is increasingly recognized as an important factor in sleep and general physical health. One potential driver of such daily variations in sleep timing is different work and social obligations during the "working week" and weekends. To investigate the nature of weekday/weekend differences in the timing of sleep offset, we examined actigraphy records of 79,161 adult participants in the UK Biobank who wore an actiwatch for 1 week. The time of sleep offset was found to be on average 36 min later on weekends than on weekdays, and when this difference was expressed as an absolute value (i.e., irrespective of sleep offset being either later or earlier on weekends), it was 63 min. Younger age, more socioeconomic disadvantage, currently being in employment, being a smoker, being male, being of non-white ethnicity and later chronotype were associated with greater differences in sleep offset between weekdays and weekend days. Greater differences in sleep offset timing were associated with age-independent small differences in cardiometabolic health indicators of BMI and diastolic blood pressure, but not HbA1c or systolic blood pressure. In a subset of participants with Type 2 Diabetes Mellitus, weekday/weekend sleep offset differences were associated weakly with BMI, systolic blood pressure and physical activity. Overall, this study demonstrates potentially substantive differences in sleep offset timings between weekdays and weekends in a large sample of UK adults, and that such differences may have public health implications.

**Keywords:** sleep; sleep timing variability; social jetlag; weekday; weekend





## 1. Introduction

Day-to-day variability in sleep timing, duration and quality is increasingly being recognized as an important factor that influences health and quality of life [1]. One operationalization of daily variability in sleep timing is social jetlag, defined as the difference in the timing of midsleep between days with "work" commitments and "free" days without such commitments [2]. Social jetlag is reported to be prevalent in working-age adult populations [3,4], with around 80% of working-age adults using an alarm clock for wakening on workdays due to the mismatch between social schedules and imperatives and internal biological time [5]. At least one hour of SJL is reported in 69% of working-age adults, while over 2 h is reported in 33% [5]. Hashizaki et al. [6] noted a 40 min delay in midsleep time, a 26 min delay in bedtime and a 53 min delay in wake time on weekend days compared to weekdays, further indicating that social/work obligations are key constraints on sleep timing. Jonasdottir et al. [7] have reported that sleep onset and offset times advance with increasing age, with the most rapid changes occurring from 55 to 67 years, overlapping with typical retirement age, further indicating the role of work commitments in driving social jetlag. Such conclusions are also supported by changes in sleep timing (e.g., less SJL, later wake times on workdays) that accompanied changes in working practices during "lockdowns" in the early phase of the COVID-19 pandemic [4,8,9].

In describing SJL, the absolute value of SJL most often used (i.e., the value of SJL irrespective of whether midsleep occurs earlier or later on free days compared to workdays [10]). However, the actual difference, which includes negative values when sleep timing is earlier on "free" days than on "work" days, may also be informative in understanding sleep timing variability, as the distribution of the actual difference is less skewed than the distribution of the absolute difference [2]. Negative actual SJL may suggest that social obligations outside of work are shaping sleep timing, and such effects may become more pronounced with increasing age due to a combination of age-related shifts towards morningness and changes in work and social commitments associated with retirement [2,10]. There are currently relatively few descriptions of negative actual SJL, and it is unclear if positive and negative SJL are associated with similar effects on physical and psychological health [11]. A recent cross-sectional Japanese study of SJL reported that only 6% of the general population displayed negative SJL (3% of people in their 20s and 8.6% of people in their 60s; [12]), whilst McMahon et al. [13] reported that 14.3% of 21–35-year-olds showed negative SJL.

Depending on study design, information on working and other social schedules is not always available, and previous research has used weekdays to signify workdays and weekend days to signify work-free days in these circumstances [14]. The justification for such a pragmatic approach is based on the fact that approximately 75% of the US and European working population attend their place of work Monday through Friday [15,16]. The aim of the current study was to describe the distribution of both actual and absolute weekday–weekend differences in the timing of sleep offset in ~80,000 participants in the UK Biobank study who wore an actiwatch for 7 days. The second aim of the study was to explore associations between demographic, behavioural and metabolic health-related outcomes and weekday/weekend sleep offset differences to reveal factors that may be drivers and/or consequences of greater differences in sleep timing between weekdays and weekends.

## 2. Results

### 2.1. Demographics and Participant Descriptive Statistics

Demographics and descriptive statistics of the 79,161 participants included in the analysis are detailed in Table 1. In brief, 57.3% of participants were female, 97.3% were white, and the average age was 56.55 years (SD = 7.79). The average BMI was 26.61 (SD = 4.46), 3.4% of participants had a diagnosis of diabetes, and 93.6% identified as non-smokers. A total of 41.3% of participants were not currently in employment, and those currently working were significantly younger (M = 53.0 years, SE = 0.032) than those who were not currently in employment (M = 61.6 years, SE = 0.033, $p < 0.001$). The average self-reported sleep duration was 7:11 h (SD = 0:58), and 25.5% were morning types, 38.2% more morning than evening, 27.4% more evening than morning and 8.8% evening types.

**Table 1.** Demographics, Health, and Sleep Characteristics of the Study Sample.

|  | Sample Size | % or Mean (SD) |
|---|---|---|
| **Sociodemographic variables** |  |  |
| *Age (years)* | 79,161 | 56.55 (7.79) |
| *Sex* | 79,161 |  |
| **Female** | 45,353 | 57.3% |
| **Male** | 33,808 | 42.7% |
| *Townsend Deprivation Score* | 79,072 | −1.82 (2.77) |
| **Quintile 1** | 15,859 | −4.83 (0.55) |
| **Quintile 2** | 15,767 | −3.59 (0.30) |
| **Quintile 3** | 15,813 | −2.50 (0.34) |
| **Quintile 4** | 15,819 | −0.87 (0.64) |
| **Quintile 5** | 15,814 | 2.71 (1.82) |

**Table 1.** *Cont.*

| | Sample Size | % or Mean (SD) |
|---|---|---|
| *Ethnicity* | 78,944 | |
| **White** | 76,840 | 97.3% |
| **Mixed** | 391 | 0.5% |
| **Asian** | 657 | 0.8% |
| **Black** | 507 | 0.6% |
| **Chinese** | 174 | 0.2% |
| **Other** | 375 | 0.5% |
| *Work Status* | 79,161 | |
| **Working** | 46,438 | 58.7% |
| **Not working** | 32,723 | 41.3% |
| **Health-related Variables** | | |
| *BMI (kg/m$^2$)* | 79,161 | 26.61 (4.46) |
| *Smoker:* | 78,991 | |
| **Yes** | 5037 | 6.4% |
| **No** | 73,954 | 93.6% |
| *Alcohol:* | 79,128 | |
| **Never** | 4304 | 5.4 |
| **Special occasions only** | 7297 | 9.2 |
| **1–3 times pm** | 8377 | 10.6 |
| **1–2 times pw** | 19,669 | 24.9 |
| **3–4 times pw** | 20,936 | 26.5 |
| **Daily/Almost daily** | 18,545 | 23.4 |
| *Physical Activity(MET/h)* | 77,576 | 35.91 (35.66) |
| *Sedentary time (h/week)* | 79,105 | 4.89 (2.13) |
| *Diabetes mellitus Disgnosis* | 2697 | 3.4% |
| *HbA1c (mmol/mol)* | 74,106 | 35.39 (5.50) |
| *Systolic blood pressure (mm hg)* | 75,537 | 138.76 (19.34) |
| *Diastolic blood pressure (mm hg)* | 75,538 | 81.58 (10.56) |
| **Sleep-related variables** | | |
| *Sleep Offest on Weekdays (hh:mm)* | 79,161 | 07:03 (01:02) |
| *Sleep Offset on Weekends (hh:mm)* | 79,161 | 07:37 (01:11) |
| *Actual Weekday-Weekend Difference (h:mm)* | 79,161 | 00:34 (01:13) |
| *Absolute Weekday-Weekend Difference (h:mm)* | 79,161 | 1:03 (00:50) |
| *Self-Reported Sleep Duration (h:mm)* | 78,992 | 7:11 (00:58) |

## 2.2. Weekday and Weekend Sleep Offset Differential

The mean clock time of sleep offset determined by actigraphy during the week was 07:03 (SD = 01:03), and during the weekend it was 07:36 (SD = 01:11 min; Figure 1). The mean absolute and actual differences between sleep offset on weekdays versus weekend days were 1:03 h (range 0 to 4:26 h, SD = 0:50 h) and 0:34 h (range −3:18 h to 4:26 h, SD = 1:13 h), respectively (Figure 1).

Males had greater absolute and actual WD/WE sleep offset differences than females (mean ± SEM; 1:04 h ± 17 s vs. 1:02 h ± 14 s, $p < 0.001$, Cohen's d = 0.04 for absolute and 0:35 h ± 24 s vs. 0:33 h ± 20 s, $p < 0.001$, Cohen's d = 0.02 for actual; Figure 2A). Age was also associated with the WD/WE sleep offset differential, with younger participants (40–49 years) showing greater absolute differences than older groups (50–59 years and 60–70 years; 1:19 h ± 26 s vs. 1:04 h ± 18 s and 0:53 h ± 14 s $p < 0.001$, η2 = 0.04; Figure 2B). Those aged 50–59 years also displayed a greater absolute WD/WE sleep offset differential than those aged 60–69 years. Younger participants (40–49 years) also showed greater actual differences than older groups (50–59 years and 60–70 years; 1:00 h ± 0.35 s vs. 0:37 h ± 26 s and 0:17 h ± 22 s, $p < 0.001$, η2 = 0.05; Figure 2B). Participants not in employment had significantly lower absolute and actual WD/WE differences compared to those working (0:54 h ± 14 s vs. 1:09 h ± 14 s, $p < 0.001$, Cohen's d = 0.31 for absolute and 0:18 h ± 22 s vs. 0:45 h ± 21 s, $p < 0.001$, Cohen's d = 0.38 for actual WD/WE differences; Figure 2C).

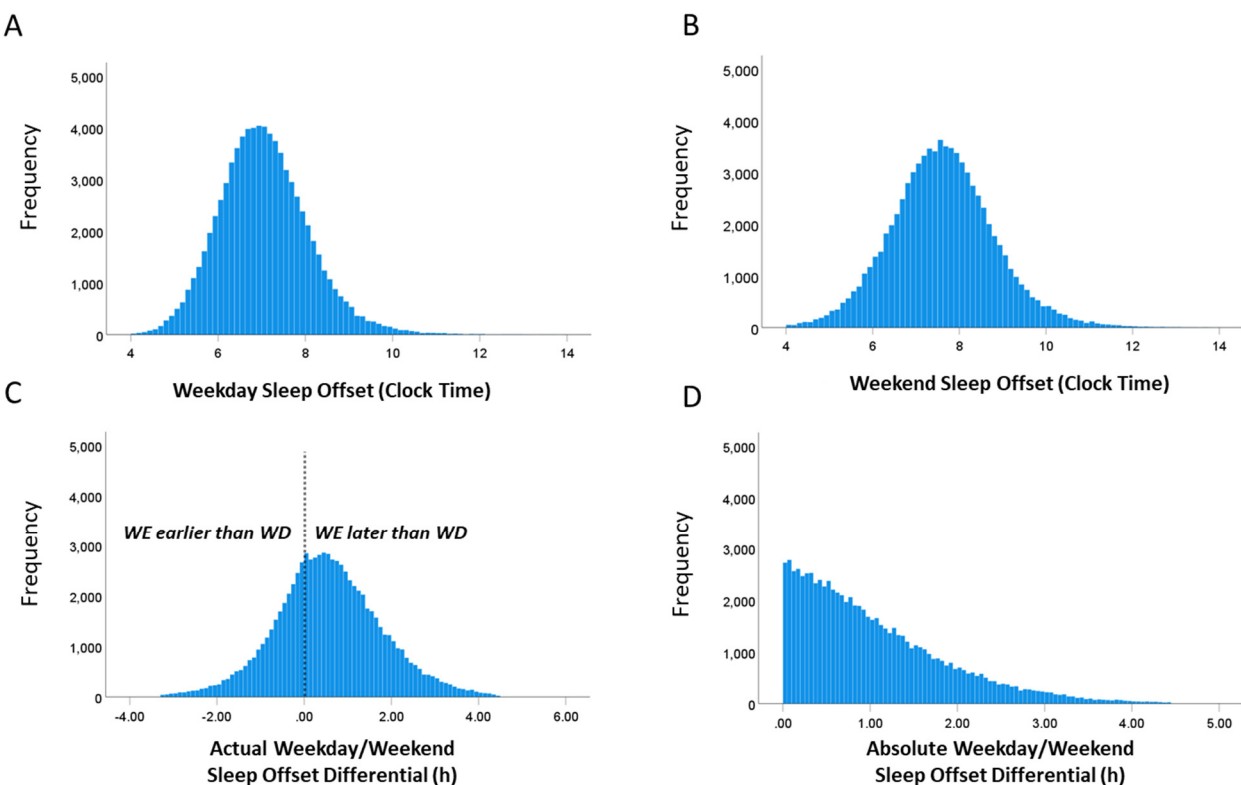

**Figure 1.** Histograms showing the distributions of (**A**) timing of sleep offset on weekdays, (**B**) timing of sleep offset on weekends, (**C**) actual weekday/weekend sleep offset differentials (negative values indicate that sleep offset timing is earlier on weekends than on weekdays), and (**D**) the weekday/weekend sleep offset differential expressed as absolute values.

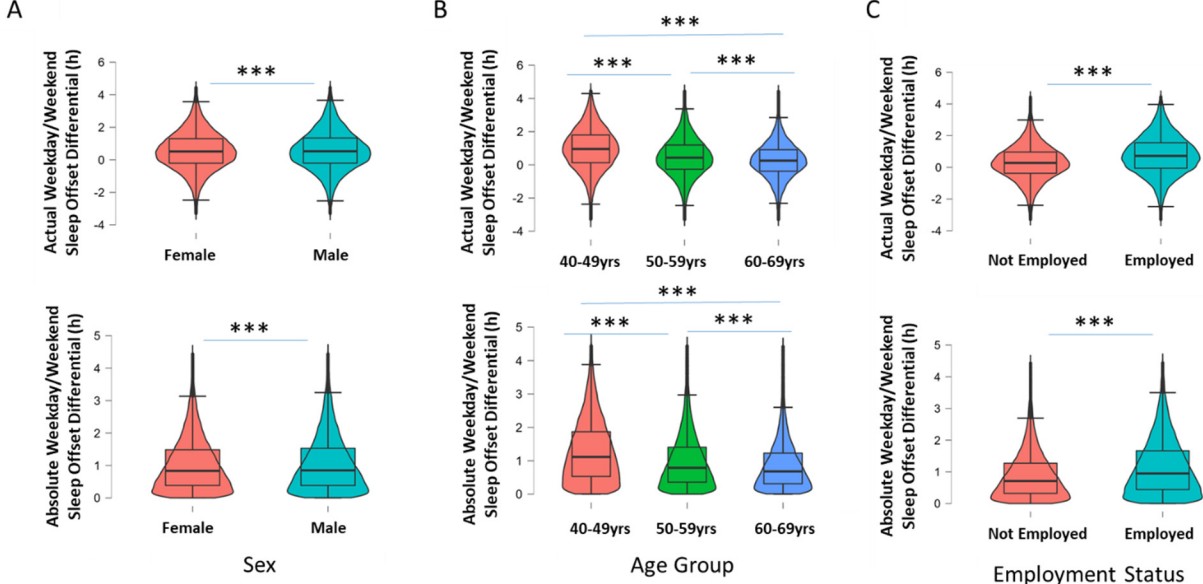

**Figure 2.** Box-and-violin plots showing (**A**) actual and absolute WD/WE in males and females; (**B**) actual and absolute WD/WE according to three age groups of participants (40 to 49 years old, 50 to 59 years old and 60 to 69 years old); and (**C**) actual and absolute WD/WE in participants currently in employment and currently not in employment. *** denotes $p < 0.001$ for pairwise comparisons (Tukey post hoc test following $p < 0.001$ for one-way ANOVA for age groups).

Since both age and employment status appear to play role in WD/WE sleep offset differentials, and as age is associated with employment status, a $2 \times 3$ between-groups ANOVA was conducted for actual WD/WE sleep offset differences by employment status (two levels) and age group (three levels; 40–49 years, 50–59 years and 60–69 years). A significant interaction was observed between age group and work status on actual WD/WE difference ($F_{(2, 79,155)} = 46.28$, $p < 0.001$, $\eta p2 = 0.20$; Figure 3). To further investigate the nature of this interaction, a number of one-way ANOVAs were run; a significant main effect of age on WD/WE sleep offset difference was found in those not currently in employment ($F_{(2, 32,720)} = 150.89$, $p < 0.001$, $\eta2 = 0.009$), as well as those currently in employment ($F_{(2, 46,435)} = 858.29$, $p < 0.001$, $\eta2 = 0.036$). The same analysis was conducted for absolute WD/WE sleep offset differences, with a significant age $\times$ employment interaction being observed ($F_{(2, 79,155)} = 14.66$, $p < 0.001$, $\eta p2 < 0.001$; Figure 2B), with a significant main effect of age on those not currently in employment ($F_{(2, 32,720)} = 206.70$, $p < 0.001$, $\eta2 = 0.012$), and those in employment ($F_{(2, 46,435)} = 636.80$, $p < 0.001$, $\eta2 = 0.027$). As such, for both measures of WD/WE sleep offset difference, there was an age-related decline in both employment status groups.

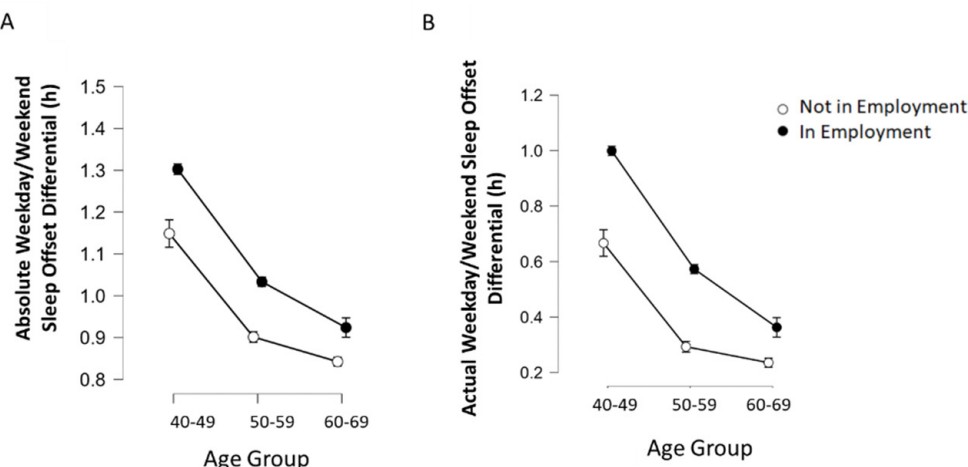

**Figure 3.** Absolute (**A**) and actual (**B**) WD/WE in participants currently in employment or not in employment, stratified by age group. There were effects of both employment status and age on both measures of WD/WE, and significant interactions between the two factors on WD/WE. Error bars (where visible) represent 95% confidence interval.

Next, the associations of self-reported chronotype, sleep duration and BMI grouping on the distribution of WD/WE differences in sleep offset timing were examined. Participants with an evening chronotype had the greatest absolute and actual WD/WE difference in sleep offsets ($F_{(3, 70,666)} = 104.8$, $p < 0.001$, $\eta2 = 0.004$ for absolute values and $F_{(3, 70,666)} = 29.68$, $p < 0.001$, $\eta2 = 0.001$ for actual values; Figure 4A). There was an effect of self-reported sleep duration grouping (<7 h, 7–8 h, >8 h; $F_{(2, 78,989)} = 16.51$, $p < 0.001$, $\eta2 < 0.001$); participants who slept less than 7 h a night had the greatest absolute WD/WE sleep offset difference (1:05 h $\pm$ 24 s) in comparison to those who slept 7–8 h (1:02 h $\pm$ 12 s) and more than 8 h (1:01 h $\pm$ 41 s Figure 4B). Similar results were observed for actual WD/WE sleep offset difference by sleep duration group ($F_{(2, 78,989)} = 47.64$, $p < 0.001$, $\eta2 = 0.001$; Figure 4B); participants who slept less than 7 h had the greatest WD/WE sleep offset difference (0:36 h $\pm$ 22 s), compared to those with 7–8 h (0:34 h $\pm$ 18 s) and more than 8 h (0:25 h $\pm$ 13 s). Obese participants (BMI of 30 or greater) had greater WD/WE sleep offset differences than overweight and normal-weight participants only when it was expressed in absolute, and not actual, terms (1:06 h $\pm$ 26 s vs. 1:02 h $\pm$ 17 s vs. 1:01 h $\pm$ 17 s); $F_{(2, 78,718)} = 43.81$, $p < 0.001$, $\eta2 = 0.001$ for actual WD/WE sleep offset difference by BMI group; Figure 4C). No significant difference was observed for actual WD/WE sleep offset differences, $F_{(2, 78,718)} = 3.10$, $p = 0.05$, $\eta2 < 0.001$; Figure 4C).

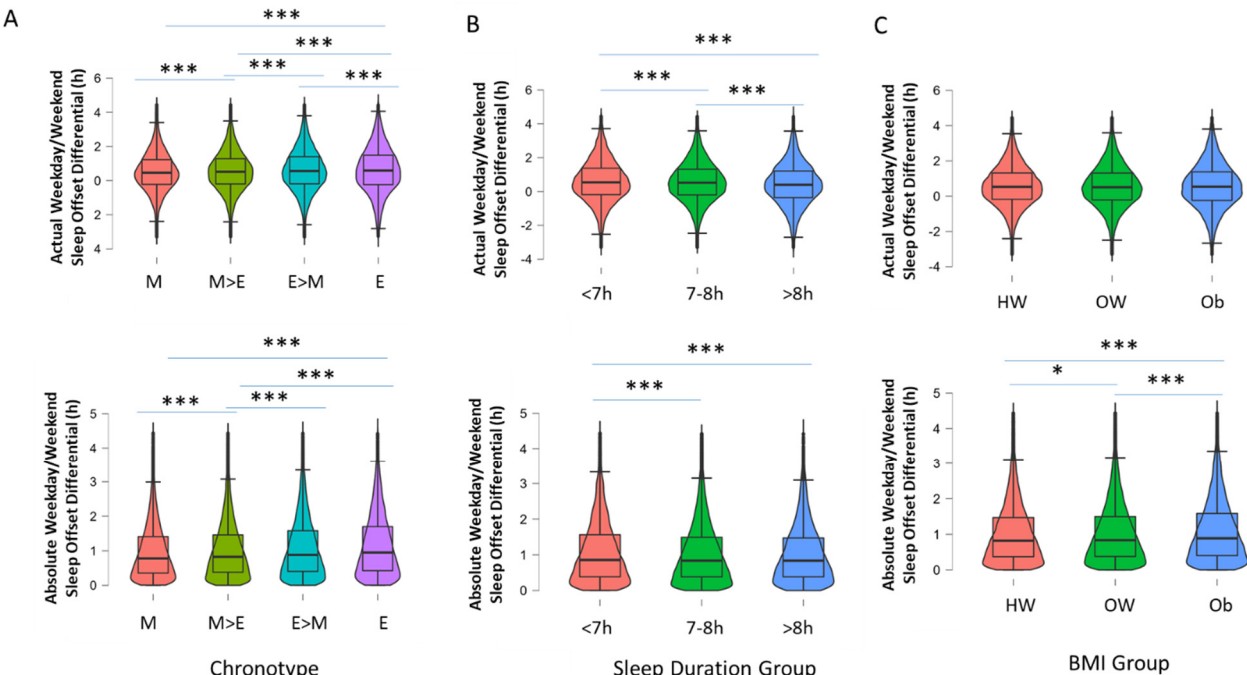

**Figure 4.** Box-and-violin plots showing (**A**) actual and absolute WD/WE according to chronotype grouping (M = morning, M > E = more morning than evening, E > M = more evening than morning and E = evening); (**B**) actual and absolute WD/WE according to three groups based on self-reported typical sleep duration (<7 h a night, 7–8 h a night, >8 h a night); and (**C**) actual and absolute WD/WE in groups based on BMI (HW = healthy weight, BMI > 19.99 < 24.99, OW = overweight, BMI > 24.99 < 29.99, OB = obesity, BMI > 29.99). *** denotes $p < 0.001$, * $p < 0.05$ for pairwise comparisons (Tukey post hoc test following $p < 0.001$ for one-way ANOVA).

Further examination of demographic variables revealed other associations with the timing of weekday/weekend sleep. Upon examining socioeconomic status via the Townsend Deprivation Index, participants in the most deprived quintile were found to have experienced a greater absolute WD/WE sleep offset difference compared to participants in the three least deprived quintiles (F(4, 79,067) = 43.91, $p < 0.001$, η2 = 0.002; Figure 5). Similarly, the actual WD/WE sleep offset difference in the most deprived participant group (0:38 h ± 20 s) was greater than that in the three least deprived quintiles (F(4, 79,067) = 19.19, $p < 0.001$, η2 = 0.001; Figure 5). As man-made artificial environmental light at night (LAN) may impact on sleep timing, we examined the association of a measure of LAN derived from satellite imaging with weekday/weekend sleep timing differences. Participants whose residences were in areas of high light at night experienced greater absolute WD/WE differences than those with lower LAN (1:05 h ± 0:18 m vs. 1:01 h ± 76 s; $p < 0.001$, Cohen's d = 0.08) and actual WD/WE differences (0:37 h ± 26 s vs. 0:32 h ± 20 s, $p < 0.001$, Cohen's d = 0.06; Figure 5). Participants who were smokers had greater absolute WD/WE sleep offset differences than non-smokers (1:09 h ± 46 s vs. 1:02 h ± 11 s, $p < 0.001$, Cohen's d = 0.14) and greater actual WD/WE sleep offset differences (0:38 h ± 67 s vs. 0:33 h ± 16 s, $p < 0.001$, Cohen's d = 0.06). Participants of non-white ethnicity also had greater levels of absolute and actual WD/WE sleep offset differences (1:13 h ± 73 s vs. 1:02 h ± 11 s, $p < 0.001$, Cohen's d = 0.21 for absolute difference; 0:45 h ± 105 s vs. 0:33 h ± 16 s, $p < 0.001$, Cohen's d = 0.16 for actual difference). Of note here, participants who resided in the highest LAN areas and those that smoked had significantly higher deprivation scores than those who did not ($p < 0.001$ for both, data not shown).

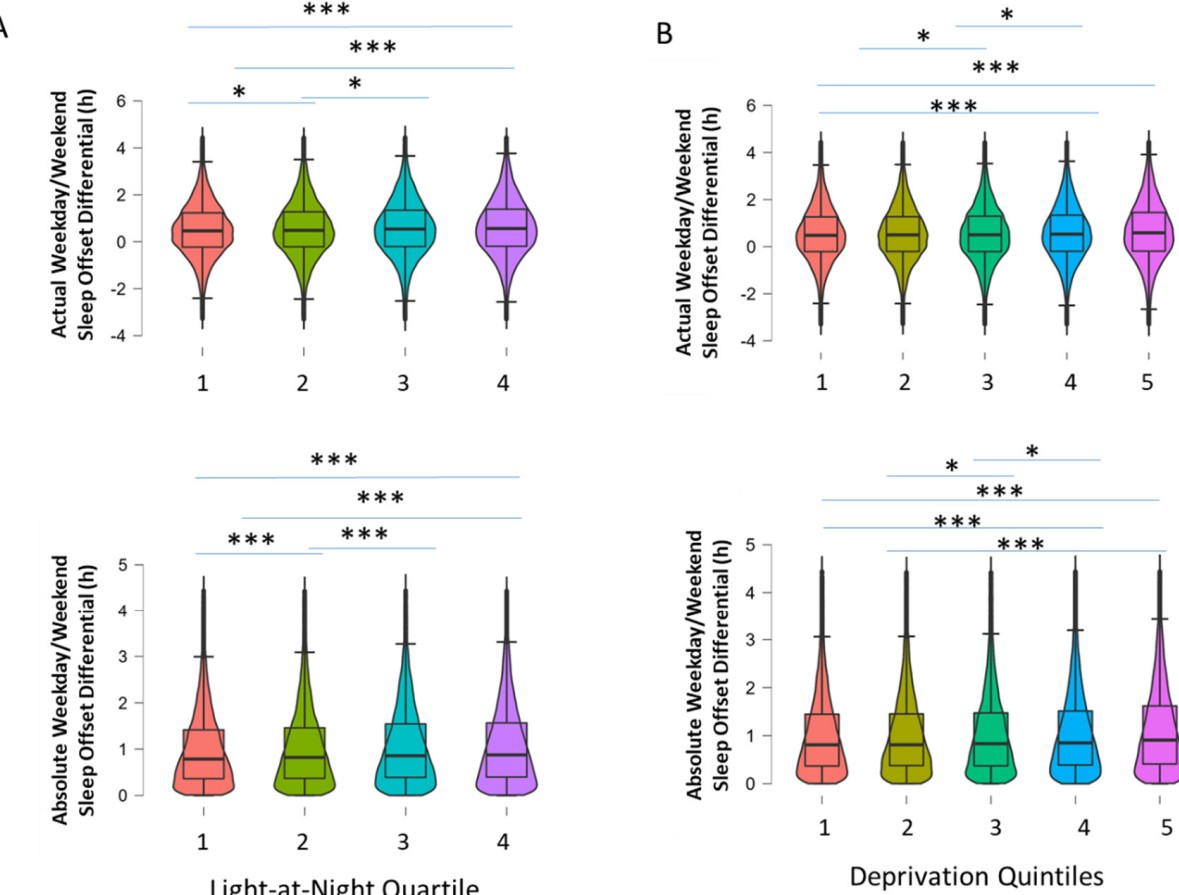

**Figure 5.** Box-and-violin plots showing (**A**) actual and absolute WD/WE according to light-at-night quartile grouping (1 = lowest LAN, 4 = highest LAN); (**B**) actual and absolute WD/WE according to quintiles based on Townsend Deprivation Index scores (1 = least disadvantaged, 5 = most disadvantaged). *** denotes $p < 0.001$, * $p < 0.05$ for pairwise comparisons (Tukey post hoc test following $p < 0.001$ for one-way ANOVA).

When the actual WD/WE sleep offset differential was categorized as a five-level variable ($>-1$ h, $-1$ h–$0$ h, $0$–$1$ h, $1$ h–$2$ h, $>2$ h), the nature of the relationships with categorical demographic and other variables was further illustrated. LAN category, smoking status, employment status, ethnicity, obesity grouping and chronotype all showed statistically significant associations with actual WD/WE grouping ($p < 0.001$ for all association via chi-square tests; Figure 6). Participants who experienced $>2$ h of WD/WE sleep offset difference were more likely to experience high levels of LAN, be smokers, be in current employment, be of non-white ethnicity, be obese and have an evening-orientated chronotype.

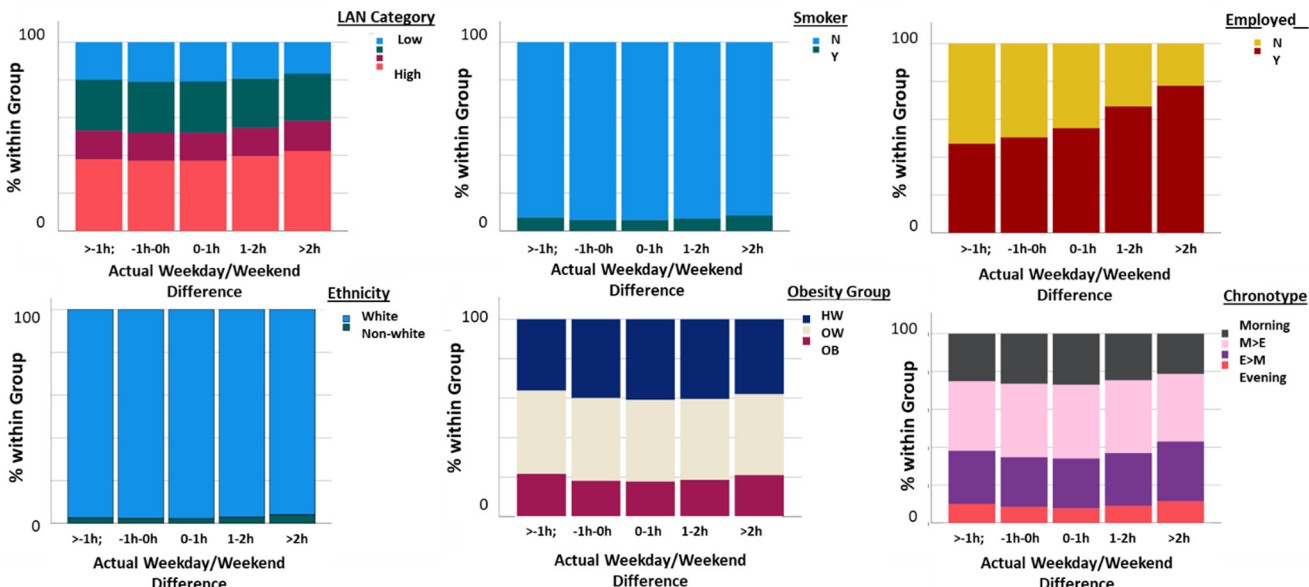

**Figure 6.** Bar graphs showing the associations between actual WD/WE sleep offset differences, expressed as a categorical variable with five levels (>−1 h, −1 h to 0 h, 0 h to 1 h, 1 h to 2 h, >2 h) with LAN categories (expressed as quartiles, low to high), smoking status (note the overall low level of smoking in the sample), employment status, ethnicity expressed as white/non-white (note the small percentage of non-white participants in the sample overall), obesity group (HW = healthy weight, OW = overweight, OB = obese) and chronotype as categorized as morning types, more morning than evening type (M > E), more evening than morning types (E > M) and evening types.

*2.3. Association of WD/WE Sleep Offset Difference with Cardiometabolic Health*

For the analysis of associations between WD/WE sleep offset differences and cardiometabolic outcomes, we stratified participants by age (40–49 years, 50–59 years and 60–69 years) as the cardiometabolic outcomes examined are markedly influenced by aging. Two-way ANOVA revealed no significant interaction between age groups and actual WD/WE sleep offset group on BMI ($F_{(8, 79,146)}$= 2.36, $p = 0.016$, $\eta p2 < 0.001$), although there was a main effect of WD/WE sleep offset differential group ($F_{(4, 79,146)} = 41.22$, $p < 0.001$, $\eta p2 = 0.002$; Figure 7A) and age on BMI ($F_{(2, 79,146)} = 123.27$, $p < 0.001$, $np2 = 0.003$). Two-way ANOVA revealed no significant interaction between age group and actual WD/WE sleep offset differential for HbA1c ($F_{(8, 74,091)} = 0.67$, $p = 0.72$, $\eta p2 < 0.001$), but there were main effects of age ($F_{(2, 74,091)} = 1220.49$, $p < 0.001$, $\eta p2 = 0.032$) and actual WD/WE sleep offset differential group ($F_{(4, 74,091)} = 4.18$, $p = 0.002$, $\eta p2 < 0.001$; Figure 7B). There was no significant interaction between age group and WD/WE sleep offset differential group observed on sedentary behaviour ($F_{(8, 77,561)} = 0.672$, $p = 0.699$, $\eta p2 < 0.001$), but there was a main effect of WD/WE sleep offset difference group ($F_{(4, 77,561)} = 11.01$, $p < 0.001$, $\eta p2 < 0.001$) and age group ($F_{(4, 77,561)} = 150.5$, $p < 0.001$, $\eta p2 < 0.004$; Figure 7C). Two-way ANOVA revealed no significant interaction between age group and actual WD/WE sleep offset differential group on systolic blood pressure ($F_{(8, 74,522)} = 1.23$, $p = 0.274$, $\eta p2 < 0.001$), and no main effect of WD/WE differential group ($F_{(4, 74,522)} = 2.25$, $p = 0.061$) but a main effect of age ($F_{(2, 74,522)} = 2668.71$, $p < 0.001$, $\eta p2 = 0.067$; Figure 7D). For diastolic blood pressure, there was no significant interaction between age group and actual WD/WE sleep offset differential group ($F_{(8, 74,523)} = 0.919$, $p = 0.499$, $\eta p2 < 0.001$), but there was a main effect WD/WE sleep offset differential group ($F_{(4, 74,523)} = 5.77$, $p < 0.001$, $\eta p2 < 0.001$) and of age ($F_{(2, 74,523)} = 196.08$, $p < 0.001$, $np2 = 0.005$; Figure 7E). No significant interaction between age group and WD/WE sleep offset differential group was observed for physical activity ($F_{(8, 77,561)} = 1.75$, $p = 0.082$, $\eta p2 < 0.001$), but there was a main effect of WD/WE sleep offset difference group ($F_{(4, 77,561)} = 4.41$, $p = 0.001$, $\eta p2 < 0.001$) and age ($F_{(2, 77,561)} = 98.11$, $p < 0.001$, $\eta p2 = 0.003$; Figure 7F). As such, these

results indicate that when there were effects of WD/WE sleep offset difference groupings on cardiometabolic outcomes, these effects were independent of age.

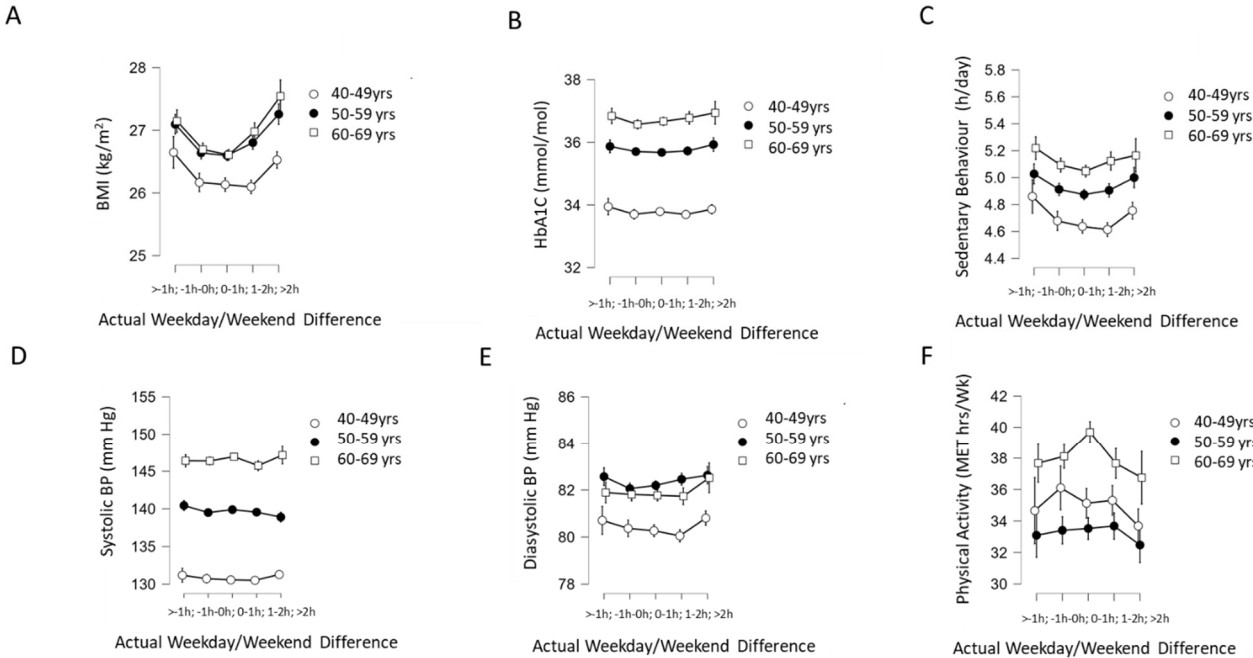

**Figure 7.** Interaction of actual WD/WE (categorized into five groups: >−1 h, −1 h to 0 h; 0 h to 1 h; 1 h to 2 h; >2 h) with age (in three groups: 40–49 years old, 50–59 years old and 60–69 years old) on (**A**) BMI, (**B**) HbA1c, (**C**) sedentary behaviour, (**D**) systolic blood pressure, (**E**) diastolic blood pressure and (**F**) physical activity expressed in metabolic equivalent of energy (MET) hours per week. For all dependent variables, there were no statistically significant interactions detected between WD/WE and age group. Error bars (where visible) indicate 95% confidence intervals.

## 2.4. Multiple Regression Analysis

In order to further examine the associations of demographic, sleep and cardiometabolic factors with WD/WE sleep offset differences, we undertook hierarchical multiple regression with absolute WD/WE sleep offset difference as the dependent variable and the sequential additional of blocks of predictor variables (age and sex in Model 1, Townsend Deprivation Index score, LAN (highest/other), smoker (yes/no) and work status (employed/not employed) added in Model 2, sleep duration and chronotype (morning/evening) added in Model 3 and BMI, HbA1c, physical activity, sedentary behaviour, systolic BP and diastolic BP added in Model 4; Table 2. For the complete model in Model 4, the $R^2$ was 0.052 and age, sex, deprivation, LAN, smoking status, work status, chronotype, BMI and physical activity were significant predictors, with age having the largest β value (β = −0.181) followed by employment status (β = 0.050), chronotype (β = 0.041) and BMI (β = 0.038; Table 3).

## 2.5. Weekday–Weekend Day Differences in People with Diabetes Mellitus

As glycaemic control in diabetes mellitus has previously been associated with social jet-lag [17], we examined WD/WE sleep offset time differences in a subset of 1691 participants who had HbA1c ≥ 42 and were classed as having a diagnosis of diabetes by running the same hierarchical multiple regression model reported above, but limited to the individuals identified in the cohort as having diabetes mellitus. In these models, only age and working status were significant predictors of absolute WD/WE sleep difference, and the addition of sleep duration, chronotype and cardiometabolic variables did not significantly increase the $R^2$ of the models.

**Table 2.** Hierarchical MultipleLlinear Regression Models with Absolute WD/WE Sleep Offset Difference as the Dependent Variable for the Full Sample.

| | $R^2$ | $R^2$ Change | β | B | SE | CI 95% (B) |
|---|---|---|---|---|---|---|
| **Model 1** | 0.045 *** | | | | | |
| *Age* | | | −0.212 *** | −0.023 | 0.000 | −0.024/−0.022 |
| *Sex* | | | 0.035 *** | 0.060 | 0.007 | 0.047/0.073 |
| **Model 2** | 0.048 *** | 0.003 *** | | | | |
| *Age* | | | −0.181 *** | −0.019 | 0.001 | −0.020/−0.018 |
| *Sex* | | | 0.032 *** | 0.054 | 0.007 | 0.041/0.067 |
| *Deprivation* | | | 0.025 *** | 0.008 | 0.001 | 0.05/0.010 |
| *LAN* | | | 0.011 * | 0.019 | 0.008 | 0.04/0.034 |
| *Smoker* | | | 0.018 *** | 0.063 | 0.014 | 0.036/0.089 |
| *Work Status* | | | 0.050 *** | 0.085 | 0.008 | 0.069/0.101 |
| **Model 3** | 0.050 *** | 0.002 *** | | | | |
| *Age* | | | −0.177 *** | −0.019 | 0.001 | −0.020/−0.018 |
| *Sex* | | | 0.031 *** | 0.053 | 0.007 | 0.040/0.066 |
| *Deprivation* | | | 0.024 *** | 0.007 | 0.001 | 0.005/0.010 |
| *LAN* | | | 0.010 * | 0.017 | 0.008 | 0.003/0.032 |
| *Smoker* | | | 0.016 *** | 0.053 | 0.014 | 0.026/0.080 |
| *Work Status* | | | 0.051 *** | 0.086 | 0.008 | 0.070/0.102 |
| *Sleep Duration* | | | −0.005 | −0.004 | 0.003 | −0.011/0.003 |
| *Chronotype* | | | 0.043 *** | 0.074 | 0.007 | 0.061/0.088 |
| **Model 4** | 0.052 *** | 0.002 ** | | | | |
| *Age* | | | −0.181 *** | −0.019 | 0.001 | −0.021/−0.018 |
| *Sex* | | | 0.027 *** | 0.047 | 0.007 | 0.018/0.062 |
| *Deprivation* | | | 0.020 *** | 0.006 | 0.001 | 0.003/0.008 |
| *LAN* | | | 0.011 * | 0.018 | 0.008 | 0.03/0.033 |
| *Smoker* | | | 0.015 *** | 0.051 | 0.013 | 0.026/0.080 |
| *Work Status* | | | 0.050 *** | 0.085 | 0.008 | 0.069/0.101 |
| *Sleep Duration* | | | −0.006 | −0.005 | 0.003 | −0.012/0.01 |
| *Chronotype* | | | 0.041 *** | 0.071 | 0.007 | 0.057/0.084 |
| *BMI* | | | 0.038 *** | 0.007 | 0.001 | 0.006/0.009 |
| *HbA1c* | | | 0.005 | 0.001 | 0.001 | −0.000/0.002 |
| *Sedentary Behaviour* | | | 0.11 ** | 0.004 | 0.002 | 0.001/0.007 |
| *Physical Activity* | | | −0.006 | −0.000 | 0.000 | −0.000/−0.000 |
| *SBP* | | | −0.004 | −0.000 | 0.000 | −0.001/0.000 |
| *DBP* | | | 0.001 | 0.000 | 0.000 | −0.001/0.001 |

*Note:* β = standardized beta value; B = unstandardized beta value; SE = Standard errors of B; CI 95% (B) = 95% confidence interval for B; N=60,710; Statistical significance: * $p < 0.05$; ** $p < 0.01$; *** $p < 0.001$. LAN = high light at night vs low; Chronotype =Morning vs evening; SBP=systolic blood pressure; DBP=diastolic blood pressure.

**Table 3.** Hierarchical Multiple Linear Regression Models with Absolute WD/WE Sleep Offset Difference as the Dependent Variable in a subset of participants with diabetes mellitus.

| | $R^2$ | $R^2$ Change | β | B | SE | CI 95% (B) |
|---|---|---|---|---|---|---|
| **Model 1** | 0.036 *** | | | | | |
| *Age* | | | −0.191 *** | −0.024 | 0.003 | −0.031/−0.018 |
| *Sex* | | | 0.013 | 0.024 | 0.047 | −0.69/0.117 |
| **Model 2** | 0.045 *** | 0.009 * | | | | |
| *Age* | | | −0.138 *** | −0.018 | 0.004 | −0.025/−0.010 |
| *Sex* | | | 0.009 | 0.016 | 0.048 | −0.078/0.109 |
| *Deprivation* | | | 0.018 | 0.005 | 0.008 | −0.111/0.022 |
| *LAN* | | | 0.036 | 0.027 | 0.022 | −0.015/0.070 |
| *Smoker* | | | −0.008 | −0.025 | 0.083 | −0.188/0.139 |
| *Work Status* | | | 0.097 ** | 0.171 | 0.054 | 0.066/0.276 |

**Table 3.** *Cont.*

|  | R² | R² Change | β | B | SE | CI 95% (B) |
|---|---|---|---|---|---|---|
| **Model 3** | 0.046 *** | 0.002 |  |  |  |  |
| *Age* |  |  | −0.136 *** | −0.017 | 0.004 | −0.025/−0.010 |
| *Sex* |  |  | 0.010 | 0.018 | 0.048 | −0.078/0.109 |
| *Deprivation* |  |  | 0.016 | 0.005 | 0.008 | −0.112/0.021 |
| *LAN* |  |  | 0.037 | 0.028 | 0.022 | −0.015/0.070 |
| *Smoker* |  |  | −0.009 | −0.027 | 0.083 | −0.190/0.137 |
| ***Work Status*** |  |  | 0.094 ** | 0.167 | 0.054 | 0.066/0.276 |
| *Sleep Duration* |  |  | −0.023 | −0.018 | 0.021 | −0.060/0.023 |
| *Chronotype* |  |  | 0.015 | 0.026 | 0.047 | −0.066/0.119 |
| **Model 4** | 0.049 *** | 0.003 |  |  |  |  |
| *Age* |  |  | −0.127 *** | −0.017 | 0.004 | −0.025/−0.010 |
| *Sex* |  |  | 0.009 | 0.018 | 0.048 | −0.078/0.109 |
| *Deprivation* |  |  | 0.012 | 0.005 | 0.008 | −0.112/0.021 |
| *LAN* |  |  | 0.035 | 0.028 | 0.022 | −0.015/0.070 |
| *Smoker* |  |  | −0.009 | −0.027 | 0.083 | −0.19/0.137 |
| ***Work Status*** |  |  | 0.096 ** | 0.167 | 0.054 | 0.066/0.276 |
| *Sleep Duration* |  |  | 0.021 | −0.018 | 0.021 | −0.060/0.023 |
| *Chronotype* |  |  | 0.014 | 0.026 | 0.047 | −0.066/0.119 |
| *BMI* |  |  | 0.038 | 0.007 | 0.001 | 0.006/0.009 |
| *HbA1c* |  |  | −0.013 | 0.001 | 0.001 | −0.000/0.002 |
| *Sedentary Behaviour* |  |  | 0.13 | 0.004 | 0.010 | −0.001/0.001 |
| *Physical Activity* |  |  | −0.015 | 0.000 | 0.000 | −0.001/0.001 |
| *SBP* |  |  | −0.023 | 0.001 | 0.002 | −0.004/0.002 |
| *DBP* |  |  | 0.022 | 0.002 | 0.003 | −0.004/0.007 |

*Note:* β = standardized beta value; B = unstandardized beta value; SE = Standard errors of B; CI 95% (B) = 95% confidence interval for B; N=60,710; Statistical significance: * $p < 0.05$; ** $p < 0.01$; *** $p < 0.001$. LAN = high light at night vs low; Chronotype =Morning vs evening; SBP = systolic blood pressure; DBP = diastolic blood pressure.

## 3. Discussion

This study investigated actual and absolute weekday–weekend day sleep offset differences in a large dataset of adult participants in the UK Biobank cohort. Actigraphy provided a measure of actual sleep timing during weekdays and the weekend within the 7-day period the actiwatches were worn for, and as such provided new objective sleep timing data that are free from recall bias and other limitations inherent in the use of questionnaires to assess such factors [17]. The main findings of the study are that the magnitude of the difference in sleep offset timing between weekdays and weekends is between ~30 min and 1 h, depending on whether actual or absolute measures are used. Higher levels of WD/WE differences were associated with high levels of LAN, currently being a smoker, being in current employment, being of non-white ethnicity, being obese and having an evening-orientated chronotype. Age was the strongest independent predictor of WD/WE differences, followed by current employment status.

The WD/WE sleep offset difference measure used in the current study is clearly not synonymous with SJL; work schedule information was not available in the UK Biobank, and as such SJL could not be directly calculated. A recent longitudinal study used differences in sleep offset timing between week and weekend days as one measure of circadian misalignment [7]. WD/WE differences in sleep timing have also been implicated in cognitive function, and may represent an important facet of sleep timing variability in and of itself (Zhang et al., 2020). Further, as approximately 75% of working adults attend their place of work Monday through Friday [15,16], it is reasonable to assume that the WD/WE measure used in this study has a meaningful overlap with SJL. This is spoken to by the finding that employment status is associated with WD/WE differences, as retirement from work is also associated with markedly decreased SJL [10]. We also found the WD/WE sleep offset differences decreased with age, and that this age-related decrease was non-synonymous with the effect of employment status. Previous work has demonstrated that SJL is also heavily influenced by age, partially through age-related declines in the eveningness/late

chronotype [2,3,18,19]; in the current study, we also found the WD/WE sleep offset difference was influenced by chronotype, with evening chronotypes having the greatest WD/WE sleep offset differences. Paine and colleagues [20] report that weekends are associated with more delay in sleep offset than sleep onset, indicating that the sleep timing point chosen to estimate sleep timing may be important in the effect sizes observed. A recent study which developed a measure of sleep timing regularity in the UK Biobank sample (Sleep Regularity Index [21]) reported that being male, having more socioeconomic deprivation and being of non-white ethnicity were associated with more irregular sleep timing.

There are currently few descriptions of negative SJL, as most extant studies have reported associations of absolute SJL with health, behavioural and psychological factors [11]. In the current study, a sizeable proportion (31.7%) of participants displayed negative actual WD/WE sleep offset differences. This prevalence of negative WD/WE sleep offset differences is higher than that reported in previous studies assessing midsleep/SJL differences: 14.3% in a study of 21–35-year-olds conducted by McMahon et al. [13], 6% of the overall population in Komada et al. [12] and only 1.7% in Roenneberg et al. [5]. Hashizaki et al. [6] noted an average 26 min delay in bedtime and a 53 min delay in wake time between weekdays and weekend days, and younger individuals show greater weekend changes in sleep timing. These differences may be explained by a combination of differences in the age profile of the current participants (the average age in the current sample was ~56.5 years old compared to other studies whose samples were drawn from younger adults) and the non-synonymous nature of the WD/WE sleep offset measure to SJL. As shift workers were excluded from the current analysis, it is not immediately apparent what contributes to earlier wakening on weekends compared to weekdays, and future quantitative and qualitative research might usefully explore this question.

In the current study, participants with the highest deprivation scores and those exposed to the highest levels of artificial LAN reported greater absolute and actual weekday-to-weekend day differences in sleep offset timing. It is well-established that light in the evening acts to delay the circadian phase [22] and it may be through such phase delays that higher LAN results in greater actual and absolute weekday-to-weekend sleep offset differences. LAN is also reported to be associated with social deprivation, with more deprived individuals experiencing higher levels of LAN measured at the neighbourhood level (current data and [23]). Smokers also had greater actual and absolute week-to-weekend day differences in sleep offset timing, in agreement with previous work showing greater rates of smoking in people experiencing greater SJL [24]. As higher smoking rates are also linked with greater deprivation ([25] and in the current study), it is likely that there is an overlap between the association of smoking and SES on WD/WE sleep offset differences. It has been indicated that health-related behaviours may mediate the relationship between circadian disruption and adverse physical health outcomes in shift workers [26], and as such smoking status may mediate or moderate relationships between WD/WE sleep offset variability and health outcomes.

Greater SJL has previously been associated with adverse physical and psychological health outcomes, with proposed putative mechanisms involving desynchronization of components of the internal circadian pacemaking system relative to other components of the system and to behavioural and environmental cycles [27,28]. As previous work has associated cardiometabolic health with SJL [19], we investigated associations of cardiometabolic variables with WD/WE sleep offset differences and found associations of WD/WE sleep offset differences with BMI, HbA1c and diastolic blood pressure. However, these effects were small, and appeared to be independent of age (within the range examined in this study). Some previous work has failed to detect association between SJL and blood pressure [13], whilst others have reported an association [29]. There are reported associations between SJL and BMI [5,29,30] and HbA1c in type 1 and 2 diabetes patients [19]. Although the reported effect sizes are small, their potential to impact on important health variables in a substantial proportion of the general population indicates that such considerations may be of importance for public health. For the subset of participants with diabetes, regression

analysis did not reveal any associations between WD/WE sleep offset differences and any cardiometabolic measure, and as such the cardiometabolic health implications of WD/WE may be different in the context of established disease.

*Strengths and Limitations*

There are a number of strengths of the current study. Firstly, our approach utilized actigraphy to provide real-world objective measures of sleep/wake behaviour in a large cohort of participants, without the need for subjective measures and the caveats that the use of those entails [17]. Secondly, given the level of in-depth characterization of participants in the UK Biobank, we were able to explore relationships with a number of cardiometabolic measures and biomarkers as well as with demographic factors. Lastly, an important concomitant strength and caveat for analyzing such datasets is the very high statistical power that may allow for the reporting of very small effect sizes and weak associations; such results need to be carefully evaluated for meaningfulness of any statistically significant results revealed in the context of the specific measures examined.

There are some important caveats and limitations that should be considered in the interpretation of the presented data. Firstly, we focused on sleep offset timing rather than sleep onset or midsleep time, and the analysis of these measures of sleep timing may yield different results. Sleep offset timing has been used recently in a large study [7] and at present we are unaware of compelling evidence that suggests that sleep onset or midsleep timing measures would be more meaningful than sleep offset timing. However, this is clearly an issue that warrants future attention in large actigraphy datasets and investigation of other measures of sleep timing variability may yield new insights. As noted earlier, no information on the distribution of actual work and work-free days was available for the participants, and as such it is not known whether or not participants worked on the weekend and shifted their weekend sleep offset to be earlier as a result. However, the measure of WD/WE sleep offset differences does offer valuable information on how people may shift their sleep timing over the week and has been used previously as an indicator of sleep timing variability [7,14]. Another important limitation is that as per the UK Biobank study protocol, there was a delay between the gathering of baseline measurements (2006–2010) and the collection of the actigraphy data (2013–2016), and this may have produced results that would be different if all measures were collected in parallel. Actigraphy data were collected across a seven-day period, and so for each participant there is only one "weekday" and "weekend" period recorded; collection of data over a longer timeframe may reveal differences between working weeks and weekends in participants. Finally, the UK Biobank study cohort comprises middle-aged and older adults with a demographic skew towards white ethnicity, and as such may not be fully nationally or internationally representative.

## 4. Materials and Methods

### 4.1. Sample

The study sample were participants who took part in the UK Biobank study. Initially, over 500,000 participants in the UK National Health Service registry were recruited as part of this study between 2006 and 2010 (UKB handbook). A subset of these participants wore wrist activity monitors (AX3 triaxial accelerometer, Axivity, Newcastle upon Tyne, UK) for 7 days, and inclusion in this current study was restricted its analysis to participants with actigraphy data that passed quality control and contained sufficient weekend day and weekday data to allow for analysis of sleep offset differences; data pre-processing was conducted by the UK Biobank Accelerometer Expert Working Group, with details available at http://biobank.ctsu.ox.ac.uk/crystal/docs/PhysicalActivityMonitor.pdf (accessed on 1 August 2022). Further details on the quality control applied to the actigraphy data can be found in [31]. This study was covered by the generic ethical approval for UK Biobank studies from the NHS National Research Ethics Service (approval letter dated 17 June 2011, Ref 11/NW/0382) for project #26209. All participants provided their full

informed consent to participate in the UK Biobank and to have their data analyzed for all extending research. This research was further approved by the Maynooth University Research Ethics Committee.

### 4.2. Participant Measures

Participants attended 1 of the 22 assessment centres across the UK. At baseline, all participants completed numerous touchscreen questionnaires, interviews, and anthropometric assessments. Blood samples were also provided at baseline. Demographics included were age, sex (male/female) and ethnicity (White, Black, Mixed, Chinese, Asian, Other), which were self-reported with a touch-screen questionnaire. Townsend Deprivation Index score based on postcode of residence was calculated for each participant as a measure of socioeconomic status; this was split into quintiles to aid analysis. Chronotype was self-reported through individuals answering the question "Do you consider yourself to be . . . " with one of the following responses: definitely a morning person, more of a morning person than an evening person, more of an evening person than a morning person or definitely an evening person. Some individuals also reported "do not know" or "prefer not to answer", and for the purpose of this analysis, these were coded as missing responses. Usual sleep duration was also self-reported in hours per 24 h. Total physical activity measured as metabolic equivalents (MET. hours/week) was calculated from self-reported duration and intensity of usual physical activity. Sedentary behaviour was derived from the self-reported time spent watching television, driving or using a computer, and expressed as an average value of hours per day. Average alcohol intake was reported as never, special occasions only, 1–3 times per month, 1–2 times per week, 3–4 times per week and daily/almost daily. Smoking status was categorized as current smoker or non-smoker. Light at night (LAN) at home was measured as intervals of the greyscale of tiff files derived from satellite data per participant postcode, and was categorized into quartiles. Information on work status was also collected with individuals either being classed as currently working or not working (currently working encompasses those in full and part-time employment; "In Employment" refers to being in paid employment, "Not in Employment" refers to being retired, looking after home and/or family, being unable to work because of sickness/disability, being unemployed, doing unpaid or voluntary work and being a full-time student). Anthropometric measures including BMI (kg/m$^2$), and systolic and diastolic blood pressure were assessed at baseline by trained UK Biobank staff using standardized instruments and measurements. HbA1c was measured by HPLC analysis on a Bio-Rad VARIANT II Turbo from serum samples (further details at https://biobank.ndph.ox.ac.uk/ukb/ukb/docs/serum_hb1ac.pdf; accessed 1 February 2022).

### 4.3. Sleep Offset Differences between Week and Weekend Days

Sleep offset times on weekdays (Tuesday–Thursday) were subtracted from sleep offset times on weekend days (Saturday and Sunday). Sleep offset values were obtained from participants activity monitors. Participants wore an axivity AX3 wrist-worn triaxial accelerometer on their dominant hand for 7 days. These physical activity monitors were worn by a subset of participants between June 2013 and January 2016. The data this collected were analyzed using the ClockLab software (v6; Actimetrics, Wilmete, IL, USA) to derive daily times of sleep offset. The analysis of weekday–weekend day sleep offset differences was assessed as both actual and absolute differences with actual difference including negative values and the absolute difference describing values' distance from zero.

### 4.4. Data Screening and Statistical Analysis

The full sample available with a measure of weekday–weekend day sleep offset differences was 87,590. A total of 6820 participants were self-reported shift workers and were excluded from subsequent analysis. Values of less than 3 h and more than 13 h for self-reported sleep duration per night were excluded (N = 26); BMI values < 12, >60 or not reported were also excluded from our analysis (N = 206). For actual WD/WE

sleep offset differences, those outside of three standard deviations of the mean were also excluded (N = 744). Of the remaining 79,793 respondents, we removed data from a further 557 participants with no information on work status. Participants with physical activity measures more than three standard deviations away from the mean were coded as missing due to unrealistic values; 1585 were excluded for those with MET. h values below 192.6 and above 930.0. Where sample size varied for some variables, this is noted throughout in the presentation of the results.

Analysis was conducted using IBM SPSS version 25 and JASP stats. Absolute and actual weekday–weekend day sleep offset differences were primarily examined as continuous variables but categorized into a five-level independent variable for factorial ANOVA with cardiometabolic outcomes as the dependent variables. As many health variables of interest in this sample are known to be age-dependent, factorial ANOVA utilized data stratified by age (40–49 years old, n = 24,237; 50–59 years old, n = 34,090; 60–69 years old, n = 20,745). Parametric statistics were utilized throughout due to the large sample size, which buffers against deviations from normality (Blanca Mena et al., 2017). Independent samples t-tests with Cohen's d effect estimates were used when comparing two groups, one-way ANOVAs and $\eta2$ estimates of effect size were used to compare variables with more than two levels, and two-way ANOVAs and partial $\eta2$ estimates of effect size were used to assess interactions between age and WD/WE sleep difference groups. Chi-square tests of independence were used to examine relationships between categorical variables. As the analytical approach utilized inferential testing in a quasi-exploratory manner (i.e., in the absence of narrowly defined and pre-specified hypotheses but within an existing conceptual framework to explore relationships between measures of weekday/weekend sleep timing and demographic and health variables), alpha was adjusted for multiple comparisons by Bonferroni correction to account for inflation of study-wide error rate in inferential tests (alpha = 0.00125; [32]). Accordingly, and for parsimony, $p < 0.001$ was deemed as indicating statistical significance for all inferential tests used, and effect sizes were interpreted according to Cohen [33]. Hierarchal regression analysis was used to examine the relative associations of demographic, sleep and cardiometabolic factors with absolute weekday–weekend day sleep offset differences as the dependent factor, after testing of the assumptions of multicollinearity, homoscedasticity and normality of distribution of residuals.

## 5. Conclusions

The current study illustrates that sleep offset timings commonly vary between weekdays and weekends and that a significant proportion of participants displayed earlier sleep offset times on the weekend compared to weekdays. Male sex, younger age, being in current employment, being a smoker, having low socioeconomic status, being exposed to more light at night, lower daily activity and having higher BMI and later chronotype were all associated with greater WD/WE sleep offset differences. WD/WE sleep offset differences showed associations with cardiometabolic outcomes, indicating that this manifestation of sleep timing variability may be of interest for public health.

**Author Contributions:** Conceptualization, R.M.K., A.N.C. and J.H.M.; methodology, R.M.K. and A.N.C.; software, R.M.K.; validation, R.M.K.; formal analysis, R.M.K. and A.N.C.; data curation, R.M.K. and A.N.C.; writing—original draft preparation, R.M.K. and A.N.C.; writing—review and editing, R.M.K., A.N.C. and J.H.M.; visualization, R.M.K. and A.N.C.; supervision, A.N.C. and J.H.M.; project administration, R.M.K. and A.N.C. All authors have read and agreed to the published version of the manuscript.

**Funding:** This research received no external funding.

**Institutional Review Board Statement:** The study was conducted in accordance with the Declaration of Helsinki, and approved by the NHS National Research Ethics Service (approval letter dated 17th June 2011, Ref 11/NW/0382) for project #26209, and further ethical approval was granted by the Research Ethics Committee of Maynooth University.

**Informed Consent Statement:** Informed consent was obtained from all subjects involved in the study.

**Data Availability Statement:** The data used in this study were provided by the UK Biobank (https://www.ukbiobank.ac.uk/; accessed on 1 August 2022).

**Conflicts of Interest:** The authors declare no conflict of interest.

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
