# Peer review of "Differences in Sleep Offset Timing between Weekdays and Weekends in 79,161 Adult Participants in the UK Biobank"

_2624-5175, doi:10.3390/clockssleep4040050_

Round 1
Reviewer 1 Report
The authors analyzed the 1-week actigraphy records of 79,161 adult participants in the UK BioBank to calculate who measures of weekend-weekday difference in sleep offset, an absolute value (ie. irrespective of the sign of weekend-weekday difference) and the mean weekday/weekend difference. The absolute and mean values were found to reach 63 and 36 minutes, respectively. Further, they found significant associations of larger values with socioeconomic disadvantage, current employment, smoking, male gander, non-white ethnicity, and later chronotype were associated with greater differences in sleep offset between weekdays and weekend days. Health correlates of weekday/weekend difference were cardiometabolic health indicated by BMI and diastolic blood pressure, but not HbA1c and systolic blood pressure. Finally, in a subset of 1,937 participants with type 2 Diabetes Mellitus (HbA1c>41), the correlates of weekday/weekend difference were BMI, systolic blood pressure, and physical activity. The authors concluded that “overall, this study demonstrates substantive differences in sleep offset timings between weekdays and weekends in a large sample of UK adults, and that such differences may have public health implications”.
Title
To me, the title is somewhat misleading, because only one of aspects of sleep variability, weekend-weekday difference but not variability in weekend-weekday difference, was analyzed.
Abstract
“To investigate the nature of weekday/weekend differences in the timing of sleep offset” If this was an objective of the study, pls write in the conclusion of Abstract what was found to be “the nature the nature of weekday/weekend differences in the timing of sleep offset”.
The first part of the conclusion that “overall, this study demonstrates substantive differences in sleep offset timings between weekdays and weekends in a large sample of UK adults” was not supported by any results mentioned in this Abstract (in which sample the difference was not “substantive”? To me, the difference in a half of hour is “substantive”ly small).
As for the second part of the conclusion that “such differences may have public health implications”, they may or may not.
1. Introduction
“The aim of the current study was to analyse the distribution of both actual and absolute weekday-weekend differences” “The aim of … study” cannot be “to analyze”.
Was this a hypothesis-driven study? If yes, pls formulate the hypothesis of the study.
2. Results
2.1. Demographics and Participant Descriptive Statistics and Table 1. The initial data on sleep onset and offset on weekdays and weekends must be reported (while “Sleep Duration (h:mm)” might be calculated from these onsets and offsets).
Figures are of low quality.
Figures 2 and 6. SEM must be also shown for each point of the graphs instead of saying “Error bars (where visible) represent SEM”. At least for me, they are fully invisible in Figure 2. Why don’t the authors simply exclude symbols (with only line remaining)?
Other figures do not contain any information about statistical significance of the differences. They must be transformed in tables with adding statistics, significance level, etc. Although this information is given in the test of Results, it is hard to understand it due to absence of mean values and SEM.
3. Discussion
The first paragraph of Discussion must summarize the present study result, while the comparison with other studies/measures must be following this summary.
“These differences may be explained by a combination of differences in the age-profile of the current participants and the non-synonymous nature of the WD/WE sleep offset measure to SJL.” Pls clarify, because the major contributor is likely to be the difference in age and because this result was included in Conclusion.
“Greater SJL has previously been associated with adverse physical and psychological health outcomes” I do not think that the citations are correct, because a couple of available reviews of the literature (not cited in this place) provided inconclusive results. If fact, the same conclusion might be made from the results of the authors’ analyzes.
“There are a number of strengths of the current study…” The position of this paragraph might be exchanged in the previous limitation paragraph that is traditionally the last paragraph of Discussion.
Conclusion
In this publisher format, (5. Conclusions) are placed after Methods.
“The current study illustrates”. See general comment about figures.
“WD/WE sleep offset differences showed associations with cardiometabolic outcomes, indicating that this manifestation of sleep timing variability may be of public health interest” Once again, may or may not (see also the comment about the so-called “SJL”).
4. Materials and Methods
“Sleep offset times on weekdays (Tuesday – Thursday) were subtracted from sleep offset times on weekend days (Saturday and Sunday). Sleep offset values were obtained from participants activity monitors.” The initial values must be reported in Table 1, as well as sleep onset times (if sleep duration was reported in Table 1, such initial data must be also available.
“As many health variables of interest in this sample are known to be age dependent, factorial ANOVA utilized data stratified by age (40-49 years old, 50-59 years old, 60-69 years old).” The results on different aging groups are not fully reported including the sample size and the variables reported in Table 1 for the whole sample.
“Graphs were created on r studio, and raincloud plots were created by adapting code from Allen et al. [34]” Shame on them.
Author Response
Comments and Suggestions for Authors
The authors analyzed the 1-week actigraphy records of 79,161 adult participants in the UK BioBank to calculate who measures of weekend-weekday difference in sleep offset, an absolute value (ie. irrespective of the sign of weekend-weekday difference) and the mean weekday/weekend difference. The absolute and mean values were found to reach 63 and 36 minutes, respectively. Further, they found significant associations of larger values with socioeconomic disadvantage, current employment, smoking, male gander, non-white ethnicity, and later chronotype were associated with greater differences in sleep offset between weekdays and weekend days. Health correlates of weekday/weekend difference were cardiometabolic health indicated by BMI and diastolic blood pressure, but not HbA1c and systolic blood pressure. Finally, in a subset of 1,937 participants with type 2 Diabetes Mellitus (HbA1c>41), the correlates of weekday/weekend difference were BMI, systolic blood pressure, and physical activity. The authors concluded that “overall, this study demonstrates substantive differences in sleep offset timings between weekdays and weekends in a large sample of UK adults, and that such differences may have public health implications”.
Title
To me, the title is somewhat misleading, because only one of aspects of sleep variability, weekend-weekday difference but not variability in weekend-weekday difference, was analyzed.
Authors’ Response: We accept the point made about the potential ambiguity in the title of the paper. As such, we have amended the title to “Differences in Sleep Offset Timing Between Weekdays and Weekends in 79,161 Adult Participants in the UK BioBank”. We believe that this is a more direct and unambiguous title for the study and the data presented.
Abstract
“To investigate the nature of weekday/weekend differences in the timing of sleep offset” If this was an objective of the study, pls write in the conclusion of Abstract what was found to be “the nature the nature of weekday/weekend differences in the timing of sleep offset”.
Authors’ Response: We have amended the abstract to follow the noted sentence with: “The time of sleep offset was found to be on average 36 minutes later on weekends than on weekdays, and when this difference was expressed as an absolute value (ie. irrespective of sleep offset being either later or earlier on weekends), it was 63 minutes.” Hopefully is a clear description of the main descriptive finding relating to the nature of the difference in sleep offset timing on weekends compared to weekdays. (Line 17-20)
The first part of the conclusion that “overall, this study demonstrates substantive differences in sleep offset timings between weekdays and weekends in a large sample of UK adults” was not supported by any results mentioned in this Abstract (in which sample the difference was not “substantive”? To me, the difference in a half of hour is “substantive”ly small).
Authors’ Response: We have amended this statement to “potentially substantive”. We would note that the absolute difference described between weekdays and weekends in the study is over one hour, and I do think that there is a reasonable evidence-based case to be made that a difference of such magnitude is potentially substantive. We note elsewhere in the manuscript that implications of weekend/work-free day sleep timing being earlier than weekday/workday sleep timing is not well understood. As such, we feel the new suggested phrasing is suitably nuanced to both the current findings and the uncertainty of their implications. (Line 27)
As for the second part of the conclusion that “such differences may have public health implications”, they may or may not.
Authors’ Response: We feel that this sentence is appropriate, given both current concerns and uncertainties around sleep timing variability and public health.
Introduction
“The aim of the current study was to analyse the distribution of both actual and absolute weekday-weekend differences” “The aim of … study” cannot be “to analyze”.
Authors’ Response: Thank you for this correction, and we have amended the relevant sentence to: “The aim of the current study was to describe the distribution of both actual and absolute weekday-weekend differences in the timing of sleep offset...” (Line 65)
Was this a hypothesis-driven study? If yes, pls formulate the hypothesis of the study.
Authors’ response: We have noted in the Analysis section of the materials and methods that “… the analytical approach utilized inferential testing in a quasi-exploratory manner (ie. in the absence of narrowly defined and pre-specified hypotheses but within an existing conceptual framework to explore relationships between measures of weekday/weekend sleep timing and demographic and health variables)…” (Line 404-406).. As such, the study is best described as exploratory rather than hypothesis-driven. Indeed we have noted that: “The second aim of the study was to explore associations between demographic, behavioural and metabolic health-related outcomes and weekday/weekend sleep offset differences to reveal factors that may be drivers and/or consequences of greater differences in sleep timing between weekdays and weekends”. (Line 67-69).
- Results
2.1. Demographics and Participant Descriptive Statistics and Table 1. The initial data on sleep onset and offset on weekdays and weekends must be reported (while “Sleep Duration (h:mm)” might be calculated from these onsets and offsets).
Authors’ Response: We have now included in Table 1 and Figure 1 the timing of sleep offsets on weekdays and weekends. The “sleep duration” item reported is a from a single self-report item from the BioBank questionnaire. The ClockLab algorithim used to estimate sleep offset did not allow for sleep onset times to be reliably estimated, and as such we do not report sleep onset times (nor midsleep or actigraphically-estimated sleep durations)
Figures are of low quality.
Authors’ Response: We had initially used raincloud plots to present the relevant distributions of the data, but accept the comment that this presentation may not have been sufficiently clear or informative. As such, we have redone all of the graphs presented, replacing the offending raincloud plots with Box-and Violin plots, and indicating clearly on these where statistically significant pairwise differences occur. We hope the presentation is now clear and appropriate. We have also included an extra figure (Figure 1) to show histograms of the sleep offset and weekend/weekday differences.
Figures 2 and 6. SEM must be also shown for each point of the graphs instead of saying “Error bars (where visible) represent SEM”. At least for me, they are fully invisible in Figure 2. Why don’t the authors simply exclude symbols (with only line remaining)?
Authors’ Response: We have represented these figures (now Figures 3 and 7), with the error bars representing the 95% confidence interval. We believe that the presentation is now the most appropriate one for this data, allowing the reader to clearly delineate different groups. Given the large sample size, the error signal is low in most of the presented data.
Other figures do not contain any information about statistical significance of the differences. They must be transformed in tables with adding statistics, significance level, etc. Although this information is given in the test of Results, it is hard to understand it due to absence of mean values and SEM.
Authors’ Response: We have now indicated on the newly presented Box-and Violin plots where all of the statistically-significant pairwise differences occur. Hopefully such presentation now appropriately complements the description of the results in the text.
- Discussion
The first paragraph of Discussion must summarize the present study result, while the comparison with other studies/measures must be following this summary.
Authors Response: We have now amended the first paragraph of the discussion to include a brief synopsis of the main findings of the study: “The main findings of the study are that the magnitude of the difference in sleep offset timing between weekdays and weekends in between ~30 minutes and 1 hour, depending on whether actual or absolute measures are used. Higher levels of WD/WE differences were associated with high levels of LAN, currently being a smoker, being in current employment, being of non-white ethnicity, being obese and being of an evening-orientated chronotype. Age was the strongest independent predictor of WD/WE differences, followed by current employment status.”(Line 244-249)
“These differences may be explained by a combination of differences in the age-profile of the current participants and the non-synonymous nature of the WD/WE sleep offset measure to SJL.” Pls clarify, because the major contributor is likely to be the difference in age and because this result was included in Conclusion.
Authors’ Response: We have further clarified our point relating to the age-profile of the current sample compared to younger-adult samples which have been mostly used in comparator studies: “These differences may be explained by a combination of differences in the age-profile of the current participants (the average age in the current sample is ~ 56.5 years old compared to other studies whose samples have been drawn from younger adults) and the non-synonymous nature of the WD/WE sleep offset measure to SJL”(Line 276-277)
“Greater SJL has previously been associated with adverse physical and psychological health outcomes” I do not think that the citations are correct, because a couple of available reviews of the literature (not cited in this place) provided inconclusive results. If fact, the same conclusion might be made from the results of the authors’ analyzes.
Authors’ Response: We feel that the statement is appropriate and correct – numerous studies have associated social jetlag with adverse physical and psychological health outcomes. We do not infer that all studies show such associations, nor that social jetlag associates with all domains of health. Further, it is not immediately clear to us which recent reviews are alluded to here.
“There are a number of strengths of the current study…” The position of this paragraph might be exchanged in the previous limitation paragraph that is traditionally the last paragraph of Discussion.
Authors’ Response: We have now juxtaposed the strengths and limitations as suggested. (Line 309-316)
Conclusion
In this publisher format, (5. Conclusions) are placed after Methods. “The current study illustrates”. See general comment about figures.
Authors’ Response: We have now moved the conclusion to section 5 (after the methods). We trust our new presentation of the figures has addressed the concerns about the previous version’s quality.
“WD/WE sleep offset differences showed associations with cardiometabolic outcomes, indicating that this manifestation of sleep timing variability may be of public health interest” Once again, may or may not (see also the comment about the so-called “SJL”).
Authors’ Response: Indeed, we agree that there may be issues of public health interest inherent in the current findings (and may not).
- Materials and Methods
“Sleep offset times on weekdays (Tuesday – Thursday) were subtracted from sleep offset times on weekend days (Saturday and Sunday). Sleep offset values were obtained from participants activity monitors.” The initial values must be reported in Table 1, as well as sleep onset times (if sleep duration was reported in Table 1, such initial data must be also available.
Authors’ Response: As noted earlier, we now report the sleep offset timings on both weekdays and weekends in Table 1 and Figure 1, and highlight that the sleep duration scores noted as from a self-report item.
“As many health variables of interest in this sample are known to be age dependent, factorial ANOVA utilized data stratified by age (40-49 years old, 50-59 years old, 60-69 years old).” The results on different aging groups are not fully reported including the sample size and the variables reported in Table 1 for the whole sample.
Authors’ Response: We have now included the sample sizes for the age groups (age (40-49 years old, n=24,237; 50-59 years old, n=34,090; 60-69 years old, n=20,745)). We do not feel it would be informative to recapitulate the information in Table 1 according to the age groupings, as that data is already presented in graphical format in Figure 7 and the attendant text.(Line 398)
“Graphs were created on r studio, and raincloud plots were created by adapting code from Allen et al. [34]” Shame on them.
Authors’ Response: We have now removed the reference to Allen et al as we no longer present raincloud plots.
Reviewer 2 Report
Dear authors,
your article "Weekday/Weekend Variability in Sleep Timing in 79,161 Adult 2 Participants in the UK BioBank" is very interesting and important,
but:
1. you have change the order of the main parts
Introduction - Materials and Methods - Results - Discussion
2. the quality of the figure have to improve
3. language has to be checked
Author Response
Your article "Weekday/Weekend Variability in Sleep Timing in 79,161 Adult 2 Participants in the UK BioBank" is very interesting and important, but:
- you have change the order of the main parts
Introduction - Materials and Methods - Results - Discussion
Authors’ Response: We have presented the sections in the sequence outlined in the instructions for authors.
- the quality of the figure have to improve
Authors’ Response: We had initially used raincloud plots to present the relevant distributions of the data, but accept the comment that this presentation may not have been sufficiently clear or informative. As such, we have redone all of the graphs presented, replacing the offending raincloud plots with Box-and Violin plots, and indicating clearly on these where statistically significant pairwise differences occur. We hope the presentation is now clear and appropriate. We have also included an extra figure (Figure 1) to show histograms of the sleep offset and weekend/weekday differences.
- language has to be checked
Authors’ Response: We have now undertaken a careful edit of the manuscript.
Round 2
Reviewer 1 Report
The authors addressed my concerns, I have no further questions, and I support the publication of this manuscript.